# Characterizing Urban Home Gardening and Associated Factors to Shape Fruit and Vegetable Consumption among Non-Farmers in Thailand

**DOI:** 10.3390/ijerph17155400

**Published:** 2020-07-27

**Authors:** Sirinya Phulkerd, Sasinee Thapsuwan, Rossarin Soottipong Gray, Aphichat Chamratrithirong

**Affiliations:** Institute for Population and Social Research, Mahidol University, Nakhon Pathom 73170, Thailand; inksp@hotmail.com (S.T.); rossarin.gra@mahidol.ac.th (R.S.G.); aphichat.cha@mahidol.ac.th (A.C.)

**Keywords:** urbanization, home gardening, fruit and vegetable intake, lifestyle behaviors, diet, Asia

## Abstract

The purpose of this study was to investigate the association between home gardening and sufficient fruit and vegetable (FV) intake among non-farmers in Thailand, and examine the influence of socio-demographic characteristics and other associated factors on home gardening among non-farmers in urban areas. Data were collected by a cross-sectional survey of a sample of Thai non-farmers (*N* = 5634). Information on self-reported home gardening, FV intake, health-related behaviors, and socio-demographic characteristics was collected via questionnaire. The findings show that home gardening is significantly associated with sufficient FV intake among non-farmers (*p* < 0.001). Within the non-farmer group who lived in urban areas, 9% gardened FV at home. Home gardening was significantly associated with socio-demographic characteristics (sex, age and occupation), physical activity, fear of pesticide contamination of FV, and FV safety awareness among the urban non-farmers. Respondents who were female, in the middle-adulthood group, practiced regular physical activity, feared pesticide contamination, and had high awareness of FV safety had the highest probability of gardening at home (*p* < 0.05, *p* < 0.001, *p* < 0.01, *p* < 0.001 and *p* < 0.05, respectively). The Thai government should pay more attention to factors that influence urban home gardening by providing support, building local capacity, and implementing effective interventions with the urban population.

## 1. Introduction

In recent decades, the world has become increasingly urbanized, and more than half the world’s population are living in cities, including in the developing world [1]. Thailand is one developing country which is undergoing the transition from a predominantly rural to an increasingly urban society. The country has shifted from a 41% urban population in 2008 to a nearly 50% urban population in 2018 [2]. This shift is occurring not only due to migration to Bangkok, Thailand’s capital and largest city, but also due to migration to the provincial urban centers of the country. This urban transition is bringing significant challenges and changes in population lifestyles, health, and nutrition. 

While urban living can offer many opportunities, including better health care access, many urban environments may have adverse effects on lifestyles, especially in terms of poor diets. The urban food environment may offer diversity as well as greater access to unhealthy and unsafe food choices [3] such as processed foods and prepared meals, which tend to be high in fat/sugar/salt, and street food that may be inferior in nutritional quality and hygiene. This unhealthy lifestyle directly affects a person’s overall health status. Therefore, it is important for the government to redirect attention to policy and programs to promote desire for and access to higher-quality diets in the urban setting. 

Community (or home) gardening is becoming an increasingly popular and promising intervention for healthy diet promoters. Home gardening is recognized as an integral part of enhancing household and national food security in a way that supplements rations and provides essential nutrients that people may be unable to access from their food environments [4]. Home gardening, in particular, is widely adopted and practiced in various contexts, especially in locations with limited resources and institutional support in both urban and rural locales [5]. A positive relationship between gardening and psychosocial and physical health benefits is clearly evident. Community gardening has also been found to be positively associated with a population’s lifestyle behaviors and health outcomes, such as increased consumption of fruit and vegetables (FV) [6,7,8,9,10], physical activity [6,8], self-rated health [7], psychological well-being [6,10], and social interaction [6,7,10,11]. 

In spite of these data, home gardening as a health promotion strategy in low- and middle-income countries in Asia, and particularly in Thailand, has not received wide attention. In particular, urban home gardening and its influence on population diet have not been sufficiently described. Little is also known about characteristics of urban home gardeners, especially the non-farming group, who are less likely to access FV, and factors that may influence their gardening practices [6,8,12]. According to the most recent estimate, insufficient FV consumption in Thailand caused 21,650 Thai deaths in 2014 [13]. Given the challenges in meeting the recommended level of FV intake (at least 400 g per day according to the WHO [14]), creation of a healthy home food environment can be considered one of the potential solutions. Thus, it is important to identify factors which motivate urban non-farmers to practice home gardening and increase FV intake. That information would help policy makers to prioritize areas for educational messages and communication strategies to increase FV intake and improve population nutrition to reduce associated health problems in urban populations. 

The purpose of this study was to investigate the association between home gardening and sufficient FV intake among the non-farmer population in Thailand, and examine the influence of socio-demographic characteristics and other associated factors on home gardening among non-farmers in an urban area, using nationally-representative survey data. 

## 2. Materials and Methods

### 2.1. Data Collection and Study Population 

This research was conducted as part of ‘A Longitudinal Study on Fruit and Vegetable Eating Behaviors’, a nationally-representative survey of the Thai population. The sample of this study was obtained from the first wave of this longitudinal study, conducted from May to December 2018 in the four geographic regions of Thailand (Central, North, Northeast and South regions) and Bangkok. 

The survey included a representative sample of individuals aged 15 years or older. Details of the sample selection process are described in another paper (Figure 1). For the individuals aged 15 years or older, stratified multistage random sampling was adopted. Based on the database of the National Statistical Office (NSO), residents were stratified by gender, age, geographic location, and population density to ensure representativeness of the sample. The survey used the census Enumeration Area household lists provided by the NSO. Personal interviews were conducted with each participant to collect information on their socio-demographic characteristics, lifestyle behaviors, and health condition. Of 3720 households recruited in this study, 3670 successfully completed the survey. Out of the 7065 adults aged 15 years or older from 3670 households, 5634 who were non-farmers were included in the analysis for this research.

### 2.2. Variables and Measures

#### 2.2.1. Home Gardening

In this study, home gardening was used as the dependent variable. Home gardening was defined, according to Odebode (2006), as the cultivation of a small portion of land which may be around the household or within walking distance from the home [15]. This study focused on in-ground gardening, excluding small container gardening, to ensure that it can provide enough F and V for the household to eat each day. The style of home gardening could be any of the following: mixed (F and V), kitchen, backyard, farmyard or homestead garden [16,17,18,19]. The respondents were asked *‘What is your main source of FV?’* The respondents who answered that FVs they ate were mainly from gardening at home were included in the analysis. 

#### 2.2.2. Socio-Demographic and Health-Related Variables

Data on socio-demographic characteristics (including sex, age, marital status, place of residence, educational attainment, income, and occupation) were collected from sampled respondents using a structured questionnaire. 

Data were also collected on lifestyle behaviors and health condition. Smoking and alcohol drinking habits were classified as ‘never-’, ‘ex-’, and ‘current smoker/drinker’. 

Physical activity (e.g., brisk walking, running, aerobics, competitive games, and sports) was assessed in terms of frequency (days per week) and duration (minutes per day). Participants undertaking physical activity at least 30 min every day (or totaling 210 min a week of moderate intensity activity) were classified as ‘yes’ (physically active) according to previous research that recommended for the prevention of other chronic diseases [20]. The rest were classified as ‘no’ (physically less active). This variable has been used in previous studies [6,21].

The daily amount of FV consumption was categorized according to the WHO’s recommended level of FV intake for adults [14]. Respondents who reported eating at least 400 g of combined FV per day were recorded as having ‘sufficient FV intake’, while those who reported eating less than 400 g per day were recorded as having ‘insufficient FV intake’. More details on the FV intake measurement for the sample are described in an earlier study [22]. This variable was used in the National Health Examination Survey of Thailand [23].

The respondents were also asked about their knowledge, attitude, and awareness towards FV. Those who knew the daily recommended level of FV intake were recorded as ‘yes’ (having knowledge about sufficient FV intake), while the rest were classified as ‘no’ (having no knowledge about sufficient FV intake). The respondents who reasoned that they do not want to eat FV due to fear of pesticide contamination were recorded as ‘yes,’ and the rest were recorded as ‘no’. In addition, the respondents were asked to assess their awareness level of FV safety when shopping for FV by selecting one of the following options: extremely aware, moderately aware, slightly aware, or not at all aware. The respondents who reported they were extremely or moderately aware of FV safety when shopping were grouped as ‘high awareness’, while the rest who reported they were slightly or not at all aware of FV safety when shopping were grouped as ‘low awareness’.

### 2.3. Statistical Analysis

Socio-demographic characteristics and health-related behaviors were analyzed by calculating mean values, frequencies, and percentages. The association between home gardening and FV intake, and between urban home gardening and each of the independent variables was examined using analysis of variance and t-test. Binary regression analysis was used to examine the association between each independent variable with having sufficient FV intake and with practicing urban home gardening. Odds ratios (ORs) were calculated, and the threshold level of statistical significance in all analyses was a *p* value of 0.05 or less (2-tailed).

### 2.4. Ethical Approvals

This study was conducted according to the guidelines laid down in the Declaration of Helsinki and all procedures involving research study participants were approved by the Institutional Review Board (IRB) of the Institute for Population and Social Research of Mahidol University (COA. No. 2018/02-070). Written informed consent was obtained from all subjects.

## 3. Results

### Socio-Demographic Characteristics of the Sample

Table 1 presents the general characteristics of the sample. Among the 5634 respondents, 54.4% were female, with a mean age of 45.5 ± 16.5 years. Over half (57.9%) of the respondents were aged 30–59 years. Two-fifths of the respondents were unemployed, 25.7% owned their own business, and 19.4% had wage labor jobs. 

Approximately one-seventh (14.7%) of the respondents reported gardening F or V at home for their own consumption. Nearly one-third (32.1%) of the respondents had sufficient FV intake. The prevalence of sufficient FV consumption was higher in females (35.0%), aged between 30 and 44 years (36.6%), married (34.7%), living in a urban area (33.0%), obtaining at least a Bachelor’s degree (39.3%), earning monthly income lower than 10,000 baht (33.3%), having a government job (38.7%), and gardening at home (42.0%). Statistically significant differences between socio-demographic characteristics and home gardening, and sufficient FV intake were found in relation to age, education level, occupation, and practice of home gardening. 

Table 2 shows results from the binary logistic regression analysis of socio-demographic characteristics and home gardening in association with sufficient FV intake. The analysis shows that all socio-demographic variables and home gardening activity were significantly related to sufficient FV intake in the non-farmer population.

Respondents aged 30–44 years had the highest probability of having sufficient FV intake compared to other age groups (*p* < 0.001) (OR = 1.467, 95% CI 1.190–1.807). Those who were married were 1.5-fold more likely to have sufficient FV intake than those who were single (*p* < 0.001) (OR = 1.471, 95% CI 1.221–1.773). Those who worked for a company had the lowest probability of having sufficient FV intake (*p* < 0.001). The respondents who grew a FV garden at home were 1.6-fold more likely to have sufficient FV intake than those who did not have a home garden (*p* < 0.001) (OR = 1.571, 95% CI 1.341–1.840).

When considering the general and home gardening-specific characteristics of the respondents who live in an urban area, the proportion practicing home gardening was higher in females (10.0%), aged 45–59 years (11.5%), married (10.6%), not having a formal education (13.6%), earning a monthly income under 10,000 baht (11.0%), and working as a wage laborer (14.1%) (Table 3). The highest proportion of those practicing home gardening was also found in the respondents with healthy lifestyle behaviors, such as ex-smokers (13.4%), ex-alcohol drinkers (11.4%), and persons with regular physical activity (11.2%). Of all the respondents who practiced home gardening, 9.5% had knowledge about sufficient FV intake, 12.1% had fear of pesticide contamination of FV, and 9.6% had high awareness FV safety.

The results of the binary logistic regression analysis of associated factors with home gardening among urban residents found that sex, age, occupation, regular physical activity, fear of pesticide contamination of FV, and FV safety awareness were significantly related to the practice of urban home gardening (Table 4). Female adults were 1.5-fold more likely than male adults to practice home gardening (*p* < 0.05) (OR = 1.493, 95% CI 1.011–2.205). The probability of gardening at home among urban residents at age 45–59 years (*p* < 0.001) (OR = 3.182, 95% CI 1.737–5.829) was the highest compared to the other age groups. Among occupation groups, the respondents who worked for a company had the lowest probability of home gardening (*p* < 0.001) (OR = 0.213, 95% CI 0.090–0.507). 

This study also found a significant association between physical activity and urban home gardening. The respondents with regular physical activity were 1.5-fold more likely to practice home gardening (*p* < 0.01) (OR = 1.485, 95% CI 1.133–1.946) than those who had irregular physical activity. The respondents who feared pesticide contamination of FV, and had high FV safety awareness were more likely to practice home gardening (*p* < 0.001 and *p* < 0.05, respectively). After controlling the confounding effects of all the independent variables, this study did not find an association of smoking and alcohol drinking with home gardening. 

## 4. Discussion

To the best of our knowledge, this was the first study to investigate the practice of urban home gardening and associated factors among non-farmers in Thailand (or even in ASEAN-member countries). Home gardening has received growing attention as a promising strategy to improve health and nutrition, and it has been widely adopted in a variety of settings, in both rural and urban areas, and in resource-constrained settings. The prevailing literature affirms the contribution of home gardening to food and nutritional security and livelihoods via alleviating hunger and malnutrition in many developing countries [5]. 

Although non-farmers who lived in urban areas were slightly more likely to have sufficient FV intake than those who lived in rural areas, very few practiced home gardening. This research found that only a small proportion of that subpopulation grew FVs at home for their own consumption (14.7%). That low level might be due to lack of time, insufficient gardening knowledge/skills, or a disconnect with food gardening/food system. Those are the most common barriers identified among non-FV-gardeners in previous research [24,25]. That said, the results of this study did validate the significant, positive relationship between home gardening and sufficient FV intake. People who practiced home gardening were 1.6-fold more likely to have sufficient FV intake than those who did not. That finding is consistent with previous studies that found a significant relationship between home gardening and vegetable intake among Japanese persons aged 60–69 [6], and a direct influence of urban gardening on improved FV intake among older adults [7]. Therefore, home gardening should be promoted as part of people’s routine leisure activity to increase their FV access and intake, and to enhance household food security and wellbeing. 

The findings show that the practice of urban home gardening differed by socio-demographic characteristics and health-related behaviors among Thai non-farmers in urban areas. This study found a significant association of urban home gardening with sex, age, occupation, physical activity, fear of pesticide contamination of FV, and FV safety awareness. Thai urban women were more likely to garden FV at home than Thai urban men. This may be related to gender roles in Thailand where women have more household decision-making power and influence on the household’s food-related behaviors. Thai women are commonly expected to take care of household food procurement, meal preparation, and allocation, which can contribute to preserving household food culture and, ultimately, the household’s diet and nutritional status. Thus, this finding is not surprising, since previous research also shows the influence of women as the primary food providers for children and older persons in the household [26]. These findings point to the importance of women, especially in FV gardening knowledge and skills, as the key group who can positively influence nutrition and health of the household as well as their own nutrition and health. 

There was a positive association between age and urban home gardening. The highest probability of gardening at home was found among urban residents aged 45–59 years. This could be explained by the fact that middle adulthood or pre-retirement is a period of contentment and satisfaction due to job stability, financial security, and the end of child rearing, despite the physical decline that naturally occurs with aging [27]. Those ages are also a period in which the day-to-day demands of the workplace may decline, and there is more time to spend with family or at home. Accordingly, the middle-adulthood group should have more opportunity or spare time for home activity such as gardening than those in the younger age groups.

Urban people who worked in a company were least likely to garden FV at home for their own consumption compared to those working in other occupations. This finding is unsurprising as the company office is likely to be where an employee’s work and life schedule is highly influenced by their employer. Compensation and career opportunity of the employees are hugely dependent on workhours, and their overtime work is often linked to employer’s pressure to work overtime and performance [28]. Consequently, long workhours, overtime work, and work outside regular daytime affected lower balance between work and life among the company workers, having less time to leisure and less psychological energy to do things at home, such as gardening. Paradoxically, this greater devotion to the workplace can lead to occupational burnout, and that can have deleterious physical and psychological health consequences. Therefore, creating an environment in which workers can have greater control over their own work schedule, with a more rational work–life balance, should create more time for healthy leisure activity such as home gardening, and that can produce dividends for the company by creating a healthier, more productive workforce. Gardening does not only benefit the employed group, but can also be a healthy productive use of time that provides a range of psychological, physical, social and economic benefits for unemployed people.

This study found that regular physical activity was the only lifestyle behavior that was positively associated with urban home gardening. This finding is consistent with several other studies [6,8,29,30]. A previous study reported that older persons who participated in daily gardening could achieve the recommended physical activity levels (at least 30 min of moderate-intensity physical activity) [30]. Similarly, Aime et al. [29] reported higher levels of physical activity among gardeners than non-gardeners. There is incontrovertible evidence of the health benefits of regular physical activity, such as prevention of common chronic diseases and a reduced risk of premature death [31]. Thus, these results support the importance of encouraging urban residents to practice home gardening in order to increase regular physical activity, and as a means to improve their overall health. People who engage in physical activity above the recommended level are likely to gain further health benefits [31].

Those persons who did not want to eat FV due to fear of pesticide contamination and who were highly aware of FV safety were more likely to grow FV at home for their own consumption. This finding can be explained by Fear Arousal Theory, which asserts that fear is a strong emotion and a natural response to personal threats that people cannot see or do not understand [32]. Fear arousal can positively influence people’s attitudes and intentions, and then motivate them to change from life-endangering to health-promoting behaviors. Fear appeals are often used by health promoters as a method to raise awareness of risk behaviors and to change them into healthy behaviors. Therefore, using fear arousal as a communication strategy—by emphasizing the potential danger and harm from eating contaminated FV, together with recommending home-grown FV as a safe alternative—may help increase FV access and intake for urban residents.

There have not been many studies (in Thailand and ASEAN-member countries) about urban home gardening and associated factors among non-farmers, who are less likely to have access to FV compared to farmers. This study fills that gap in the literature and sheds light on a variety of socio-demographic characteristics and lifestyle behaviors as predisposing factors. In addition, this analysis had the advantage of a relatively large sample size of the urban non-farmer population compared to other studies, and that increases the robustness of the findings. However, some limitations of this study remain. Firstly, all the data in this study were self-reported, which might inevitably introduce response bias. Secondly, the data used for the analysis were from one wave of a longitudinal study. Thus, it is not possible to conclude a causal relationship between the associated factors and health-related variables. Additional research using longitudinal data is required to examine the reciprocal nature of these relationships. Further research on economic and non-economic benefits, technological support, women’s empowerment, and sustainability of urban home gardening is also needed.

## 5. Conclusions

The findings of the analysis of data from the cross-sectional study suggest that there is a significant association between home gardening and sufficient FV intake among the non-farmer population in Thailand. Findings indicate that urban home gardening might be positively affected by healthy lifestyle behavior (i.e., regular physical activity), fear of FV pesticide contamination, and FV safety awareness. Socio-demographic differences were also observed, and sex, age and occupation were associated with urban home gardening. These factors, which are modifiable to some extent, could be potential targets for increasing home gardening in the urban population. The government should pay more attention to the value and potential of home gardening and its associated factors for enhancing food security and health benefits among the urban population. The government should provide support, build local capacity, and sponsor effective interventions for urban populations to practice home gardening of FV.

## Figures and Tables

**Figure 1 ijerph-17-05400-f001:**
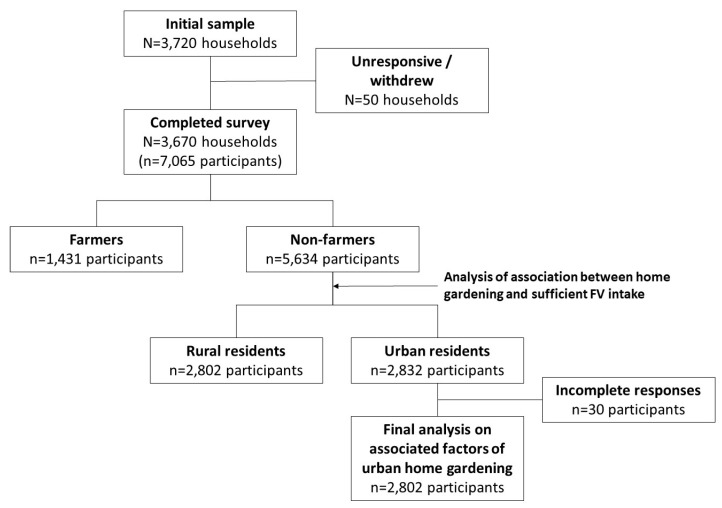
Sample selection flowchart.

**Table 1 ijerph-17-05400-t001:** Socio-demographic characteristics and home gardening for sufficient FV intake in the non-farming population (*N* = 5634).

Variables	*N*	% of Total	FV Grams/Day		*p*-Value *t*-Test
			<400	≥400	
Total	5634	100.0	67.9	32.1	-
Sex					
Male	2569	45.6	71.3	28.7	
Female	3065	54.4	65.0	35.0	0.507
Age (years)					
15–29	1207	21.4	75.5	24.5	
30–44	1360	24.2	63.4	36.6	0.006
45–59	1899	33.7	65.2	34.8	0.986
60 or over	1168	20.7	69.4	30.6	0.530
Marital status					
Single	1228	21.8	75.7	24.3	
Married	3672	65.2	65.3	34.7	0.064
Widowed/divorced/separated	734	13.0	67.8	32.2	0.691
Place of residence					
Urban	2832	50.3	67.0	33.0	
Rural	2802	49.7	68.7	31.3	0.465
Educational attainment					
No formal education	89	1.6	73.0	27.0	
Primary school	2544	45.2	68.6	31.4	0.339
Secondary school	2374	42.1	68.8	31.2	0.258
Bachelor’s or higher degree	627	11.1	60.7	39.3	0.044
Income (baht per month)					
No income	1422	25.2	70.8	29.2	
Less than 10,000	2057	36.5	66.7	33.3	0.449
10,000–19,999	1410	25.0	67.0	33.0	0.126
20,000 or above	745	13.3	67.1	32.9	0.321
Occupation					
Unemployed	2245	39.8	69.8	31.2	
Government job	313	5.6	61.3	38.7	0.978
Company hire	533	9.5	76.3	23.7	0.000
Own business	1446	25.7	63.5	36.5	0.485
Wage laborer	1097	19.4	69.5	30.5	0.296
Home gardening					
No	4808	85.3	69.9	30.4	
Yes	826	14.7	58.0	42.0	0.010

**Table 2 ijerph-17-05400-t002:** Binary logistic regression of socio-demographic characteristics and home gardening in association with sufficient FV intake (*N* = 5634).

Variables	Adjusted or (95% CI)
Sex (Reference group = Male)	
Female	1.228 (1.085–1.390) **
Age (years) (Reference group = 15–29)	
30–44	1.467 (1.190–1.807) ***
45–59	1.357 (1.092–1.687) **
60 or over	1.068 (0.830–1.375)
Marital status (Reference group = Single)	
Married	1.471 (1.221–1.773) ***
Widowed/divorced/separated	1.346 (1.050–1.724) *
Place of residence (Reference group = Rural)	
Urban	1.160 (1.031–1.305) *
Educational attainment (Reference group = No formal education)	
Primary school	1.280 (0.792–2.070)
Secondary/high school	1.586 (0.969–2.594)
Bachelor’s or higher	2.320 (1.377–3.908) **
Income (baht per month) (Reference group = No income)	
Less than 10,000	1.314 (1.081–1.596) **
10,000–19,999	1.231 (0.974–1.555)
20,000 or over	0.981 (0.749–1.285)
Occupation (Reference group = Unemployed)	
Government job	0.914 (0.669–1.249)
Company hire	0.521 (0.393–0.691) ***
Own business	0.993 (0.811–1.215)
Wage laborer	0.779 (0.630–0.963) *
Home gardening (Reference group = No)	
Yes	1.571 (1.341–1.840) ***

Note(s): Cox and Snell *R^2^* = 0.034, * Sig ≤ 0.05, ** Sig ≤ 0.01, and *** Sig ≤ 0.001.

**Table 3 ijerph-17-05400-t003:** Socio-demographic characteristics, health-related behaviors, and practicing home gardening of urban non-farming residents (*N* = 2802).

Variables	*N*	% of Total	Home Gardening	
			Yes (*N* = 252)	No (*N* = 2550)
Total	2802	100.0	9.0	91.0
Sex				
Male	1303	46.5	7.8	92.2
Female	1499	53.5	10.0	90.0
Age (years)				
15–29	563	20.1	3.5	96.5
30–44	696	24.9	8.3	91.7
45–59	1012	36.1	11.5	88.5
60 or over	531	18.9	11.1	88.9
Marital status				
Single	637	22.7	4.9	95.1
Married	1784	63.7	10.6	89.4
Widowed/divorced/separated	381	13.6	8.4	91.6
Educational attainment				
No formal education	44	1.6	13.6	86.4
Primary school	1151	41.0	10.5	89.5
Secondary school	1255	44.8	7.6	92.4
Bachelor’s or higher degree	352	12.6	8.5	91.5
Income (baht per month)				
No income	660	23.5	8.0	92.0
Less than 10,000	894	31.9	11.0	89.0
10,000–19,999	794	28.4	9.3	90.7
20,000 and above	454	16.2	5.9	94.1
Occupation				
Unemployed	1025	36.6	9.1	90.9
Government job	136	4.9	8.8	91.2
Company hire	351	12.5	2.0	98.0
Own business	772	27.5	8.7	91.3
Wage laborer	518	18.5	14.1	85.9
Smoking				
Current smoker	576	20.6	7.5	92.5
Ex-smoker	352	12.6	13.4	86.6
Never-smoker	1874	66.8	8.6	91.4
Alcohol drinking				
Current drinker	1043	37.2	8.6	91.4
Ex-drinker	446	15.9	11.4	88.6
Never-drinker	1313	46.9	8.5	91.5
Regular physical activity				
No	1587	56.6	7.4	92.6
Yes	1215	43.4	11.2	88.8
Knowledge about sufficient FV intake				
No	354	12.6	5.9	94.1
Yes	2448	87.4	9.5	90.5
Fear of pesticide contamination of FV				
No	1660	59.2	6.9	93.1
Yes	1142	40.8	12.1	87.9
FV safety awareness				
Low	411	14.7	5.8	94.2
High	2391	85.3	9.6	90.4

**Table 4 ijerph-17-05400-t004:** Binary logistic regression of socio-demographic characteristics and health-related behaviors in association with practicing urban home gardening (*N* = 2802).

Variables	Adjusted OR (95% CI)
Sex (Reference group = Male)	
Female	1.493 (1.011–2.205) *
Age (years) (Reference group = 15–29)	
30–44	2.565 (1.403–4.688) **
45–59	3.182 (1.737–5.829) ***
60 or over	2.793 (1.430–5.458) **
Marital status (Reference group = Single)	
Married	1.288 (0.806–2.058)
Widowed/divorced/separated	0.848 (0.466–1.542)
Educational attainment (Reference group = No formal education)	
Primary school	0.854 (0.340–2.148)
Secondary/high school	0.948 (0.368–2.440)
Bachelor’s or higher degree	1.447 (0.520–4.026)
Income (baht per month) (Reference group = No income)	
Less than 10,000	0.981 (0.626–1.535)
10,000–19,999	1.085 (0.650–1.809)
20,000 or over	0.649–0.345–1.220)
Occupation (Reference group = Unemployed)	
Government job	0.804 (0.372–1.736)
Company hire	0.213 (0.090–0.507) ***
Own business	0.856 (0.546–1.343)
Wage laborer	1.566 (0.998–2.456)
Smoking (Reference group = Current smoker)	
Ex-smoker	1.535 (0.955–2.467)
Never-smoker	0.990 (0.619–1.585)
Alcohol drinking (Reference group = Current drinker)	
Ex-drinker	0.958 (0.647–1.417)
Never-drinker	0.862 (0.609–1.219)
Physical activity (Reference group = No)	
Yes	1.485 (1.133–1.946) **
Knowledge about sufficient FV intake (Reference group = No)	
Yes	1.477 (0.919–2.375)
Fear of pesticide contamination of FV (Reference group = No)	
Yes	1.638 (1.250–2.147) ***
FV safety awareness (Reference group = Low)	
High	1.683 (1.077–2.630) *

Note(s): Cox and Snell *R^2^* = 0.047, * Sig ≤ 0.05, ** Sig ≤ 0.01, *** and Sig ≤ 0.001.

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
