# Peer review of "Characterizing Urban Home Gardening and Associated Factors to Shape Fruit and Vegetable Consumption among Non-Farmers in Thailand"

_ijerph, 2020, doi:10.3390/ijerph17155400_

Round 1

Reviewer 1 Report

Summary: The aims of the paper were to 1) investigate the association between home gardening and sufficient fruit and vegetable (FV) intake among the non-farmer population in Thailand, and 2) examine the influence of socio-demographic characteristics and other associated factors on home gardening among non-farmers in an urban area, using nationally-representative survey data. The authors found statistically significant differences between socio-demographic characteristics and home gardening, and sufficient FV intake in relation to age, education level, occupation, and practice of home gardening. They also observed that a low percentage of people grew FVs for their own consumption, an observation which they attribute to barriers such as lack of time, inadequate gardening skills or disconnect with gardening/food system.

Broad comments:

Strength: The article is well-written, with clear objectives and a logical flow between sections.

Weakness: Some facts in the Introduction section lack proper citation.

Specific comments:

Lines 54-58: While establishing the justification for this article, the authors stated that there is little information about characteristics of home gardeners and factors influencing their gardening practices without citing any references to buttress points raised.

Line 84: “…5,634 who were not working in farming were…” Do authors mean non-farmers?

Line 95: Is a sample of this structured questionnaire available? Was it designed specifically for this study?

Line 133: Stating only the percentage of the female study participants is good enough; readers may infer the percentage of males from Table 1.

Lines 132-147: This paragraph seemed like a repetition of Table 1. The authors may highlight the major points from Table 1 without re-stating all contents of Table 1.

Lines 150-165: Same comment as for Table 1.

Line 175-6: "... 9.5% knew about sufficient level of FV intake,..." could be replaced by "9.5% had knowledge about sufficient FV intake,..."

Line 192-3: " The adjusted analysis did not..."What variable(s) was/were adjusted for in this analysis?

Line 213: "...gardening and eating vegetables..." can be changed to "...gardening and vegetable intake/consumption..."

Line 230: Could the authors replace “association of…” with “association between…”

Lines 242-3: This sentence "...and their overtime work is often occurred due to time pressure, schedule flexibility, rewards, and pressure to work overtime" is a bit confusing to me.

Reviewer 2 Report

This study focuses on an important topic relevant to the readership of the journal. However the sample selection was not clear, a flow chart would be helpful. It appears that your independent variable, home gardening, was defined by the question, :"What is your main source of FV?" with "home gardening" defined as "home" and anything else as non home. This definition might miss-classify anyone who used home gardening as a supplement, e.g., pot gardening. Please discuss the justification for the definition of the independent variable. It also appears that your primary dependent variable, grams of FV, was defined in an unpublished article (reference 12). Please either provide a published reference or measurement details. Similarly, a secondary dependent variable was physical activity but it was unclear if this was defined as moderate to vigorous and the justification for 30 minutes a day (210 minutes/week) defining active and < 210 minutes as inactive was not clear. Please justify Tables 1 and 2 which include rural and explain why your conclusion about FV consumption was based on Table 2 including rural inhabitants as well as the lack of information about FV consumption among urban inhabitants. Finally, please provide references in lines 50-53 that are relevant to a younger population in a developed country and provide more relevant references for line 65.

Reviewer 3 Report

well organized, well written study which looks at challenges posed by continued urbanization of the human population

good return rate on surveys

line 75 please clarify/specify 'four regions of Thailand and Bankok

discussion-considering 2/5 of those surveyed were unemployed, home gardening can be a healthy productive use of time.  

Reviewer 4 Report

Your paper identified the factors associated with the practice of kitchen gardens in Thailand. This will contribute to relevant research in the future and will be of interest to the reader.
Minor comments are shown below, so please consider fixing them.

1 2.2.2. In the section (L99-116) you can cite the paper to justify the use of each variable.

2 Throughout, I used the notation "p≤". I think "p <" is more common.

3 Carefully read the guidelines in this journal and modify the reference list.
